biomathematics/computational biology/ecology

coexistence, competition, chaos, neutral competition, biodiversity paradox

**Author for correspondence:**
Pablo Rodríguez-Sánchez
e-mail: pablo.rodriguezsanchez@wur.nl

# Neutral competition boosts cycles and chaos in simulated food webs

Pablo Rodríguez-Sánchez, Egbert H. van Nes and Marten Scheffer

Department of Aquatic Ecology and Water Quality Management, Wageningen University, PO Box 47, Wageningen 6700AA, The Netherlands

PR-S, 0000-0002-2855-940X

Similarity of competitors has been proposed to facilitate coexistence of species because it slows down competitive exclusion, thus making it easier for equalizing mechanisms to maintain diverse communities. On the other hand, previous studies suggest that chaotic ecosystems can have a higher biodiversity. Here, we link these two previously unrelated findings, by analysing the dynamics of food web models. We show that near-neutrality of competition of prey, in the presence of predators, increases the chance of developing chaotic dynamics. Moreover, we confirm that chaotic dynamics correlate with a higher biodiversity.

## 1. Background

Ever since Darwin, the idea that species must be sufficiently different to coexist is deeply rooted in biological thinking. Indeed, the principles of limiting similarity [1] and competitive exclusion [2,3] are the cornerstones of ecological theory. Nevertheless, natural communities (such as plankton communities [4]), often harbour far more species that may be explained from niche separation, inspiring Hutchinson [5] to ask the simple but fundamental question *why are there so many kinds of animals?* Since then, many mechanisms have been suggested that may help similar species to coexist. As Hutchinson [4] already proposed himself, fluctuations in conditions may prevent populations from reaching equilibrium at which species would be outcompeted. Also, natural enemies including pests and parasites tend to attack the abundant species more than rare species, and such a *kill the winner* [6] mechanism promotes diversity by preventing one species from becoming dominant.

In the extensive literature on potential mechanisms that could prevent competitive exclusion, there are two relatively new ideas that have created some controversy: neutrality and chaos. The neutral theory of biodiversity introduced by Hubbell [7] proposes that species that are entirely equivalent can coexist because none

is able to outcompete the other. In the case of near-neutral communities, although *sensu stricto* the exclusion will happen eventually, the exclusion process will take a very long time to finalize. The concept of equivalent species has met scepticism, as it is incompatible with the idea that all species are different. However, it turns out that also *near-neutral* competitors can coexist in models of competition and evolution [8,9]. Support for such near-neutrality has been found in a wide range of communities [10–12]. The second controversial mechanism that may prevent competitive exclusion is *super-saturated coexistence* in communities that display chaotic dynamics [13]. This is in a sense analogous to the prevention of competitive exclusion in fluctuating environments, except that deterministic chaos is internally driven. Although there has been much debate about the question whether chaotic dynamics plays an important role in ecosystems [14–16], several studies support the idea that chaos can be an essential ingredient of natural dynamics [13,17,18].

In the present work, we used a multi-species food-web model to explore the effect of near-neutrality of prey on the probability of developing chaotic dynamics. We found a surprising link between both ideas: the closer to neutrality the competition is, the higher the chances of developing chaotic dynamics. Additionally, our results confirmed that there is a robust positive correlation between cyclic or chaotic dynamics and the number of coexisiting species.

## 2. Methods

### 2.1. Model description

We focused our attention on food webs with two trophic levels, competing prey and predators. The predators have a differentiated preference for different prey species.

The dynamics were modelled using the Rosenzweig–MacArthur predator–prey model [19], generalized to a higher number of species [20]. Our model contains $n_P$ prey species and $n_C$ predator species. The prey's populations are under the influence of both intra and interspecific competition, whose intensities are defined by the competition matrix $A$. The relative preference that predators have for each prey is defined by the predation matrix $S$. Prey immigration from neighbouring areas has been added to the classical model in order to avoid unrealistic dynamics, such as heteroclinic orbits giving rise to long-stretched cycles with near extinctions [20]. In mathematical notation, the system reads

$$\begin{cases} \frac{dP_i}{dt} = r_i(P)P_i - \sum_{j=1}^{n_C} g_j(P)P_i S_{ji}C_j + f & : i = 1..n_P \\ \frac{dC_j}{dt} = -lC_j + e\sum_{i=1}^{n_P} g_j(P)P_i S_{ji}C_j & : j = 1..n_C \end{cases} \tag{2.1}$$

where $P_i(t)$ represents the biomass of prey species $i$ at time $t$ and $C_j(t)$ the biomass of predator species $j$ at time $t$. The symbol $P$ is used as a shorthand for the vector $(P_1(t), P_2(t), \ldots, P_{n_P}(t))$. The auxiliary functions $r_i(P)$ and $g_j(P)$ (see equations (2.2) and (2.3)) have been, respectively, chosen to generalize the logistic growth and the Holling type II saturation functional response [21] to a multispecies system when inserted into equation (2.1).

$$r_i(P) = r\left(1 - \frac{1}{K}\sum_{k=1}^{n_P} A_{ik}P_k\right) \tag{2.2}$$

and

$$g_j(P) = \frac{g}{\sum_{i=1}^{n_P} S_{ji}P_i + H}. \tag{2.3}$$

For details about the parameters used, please refer to §2.2.

### 2.2. Parametrization

We parametrized our model as a freshwater plankton system based on Dakos' model [22]. Unlike Dakos, who uses seasonally changing parameters, our parameters were assumed to be independent of time (table 1).

#### 2.2.1. Competition and predation matrices

Our main purpose is to analyse the effect of different competition strengths on the long-term dynamics exhibited. For this, we introduce the competition parameter $\varepsilon$ to build a competition matrix $A$, whose

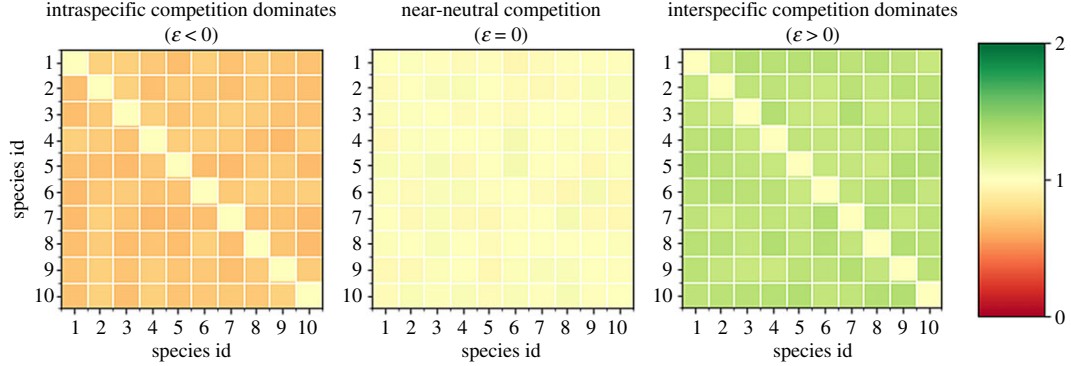

**Figure 1.** The competition matrix on the left is a clear case of dominant intraspecific competition. The central one represents a case of near-neutral competition. The matrix in the right panel shows a case of dominant interspecific competition. The difference between them is the relative size of the non-diagonal elements respective of the diagonal ones. This property of the competition matrices is controlled by the competition parameter $\varepsilon$.

**Table 1.** Values and meanings of the parameters used in our numerical experiment. The elements of the predation ($S$) and competition ($A$) matrices are drawn from probability distributions described in §2.2.1.

| symbol | interpretation | value | units |
|--------|----------------|-------|-------|
| $r$ | maximum growth rate | 0.50 | d$^{-1}$ |
| $K$ | carrying capacity | 10.00 | mg l$^{-1}$ |
| $g$ | predation rate | 0.40 | d$^{-1}$ |
| $f$ | immigration rate | $10^{-5}$ | mg l$^{-1}$ d$^{-1}$ |
| $e$ | assimilation efficiency | 0.60 | 1 |
| $H$ | saturation constant | 2.00 | mg l$^{-1}$ |
| $l$ | predator's loss rate | 0.15 | d$^{-1}$ |
| $S$ | $n_C \times n_P$ predator preference matrix | see §2.2.1 | 1 |
| $A$ | $n_P \times n_P$ competition matrix | see §2.2.1 | 1 |

non-diagonal terms are drawn from a uniform distribution centred at $1 + \varepsilon$ and with a given width (here, we chose $w = 0.1$). The diagonal terms are by definition equal to 1. Defined this way, the parameter $\varepsilon$ allows us to move continuously from strong intraspecific ($\varepsilon < 0$) to strong interspecific competition ($\varepsilon > 0$), meeting neutral-on-average competition at $\varepsilon = 0$. For the rest of this paper, we will call ecosystems with $\varepsilon = 0$ *near-neutral* (figure 1).

Regarding the predation matrix $S$, we follow [22] and draw each of its coefficients from a uniform probability distribution bounded between 0 and 1.

## 2.3. Numerical experiments

Depending on the parameters and initial conditions, our model (equation (2.1)) can have three kinds of dynamics, each of them roughly corresponding to a different kind of attractor (figure 2). In a stable point attractor, species composition is constant. The limit cycle (and limit tori) attractor corresponds to periodically (or quasi-periodically) changing species composition. The last category are chaotic attractors, where the species composition changes irregularly within bounds and there is extreme sensitivity to initial conditions.

Our target is to estimate the probability of reaching each type of attractor under different assumptions about competition. For this, we analysed 25 values of the competition parameter $\varepsilon$ (defined in §2.2.1), ranging from $\varepsilon = -0.8$ to $\varepsilon = 0.8$. The lower value was chosen to ensure that the non-diagonal competition matrix elements were positive and non-negligible to exclude facilitation and non-competing species. The upper value was arbitrarily chosen to be symmetric with the lower one. For

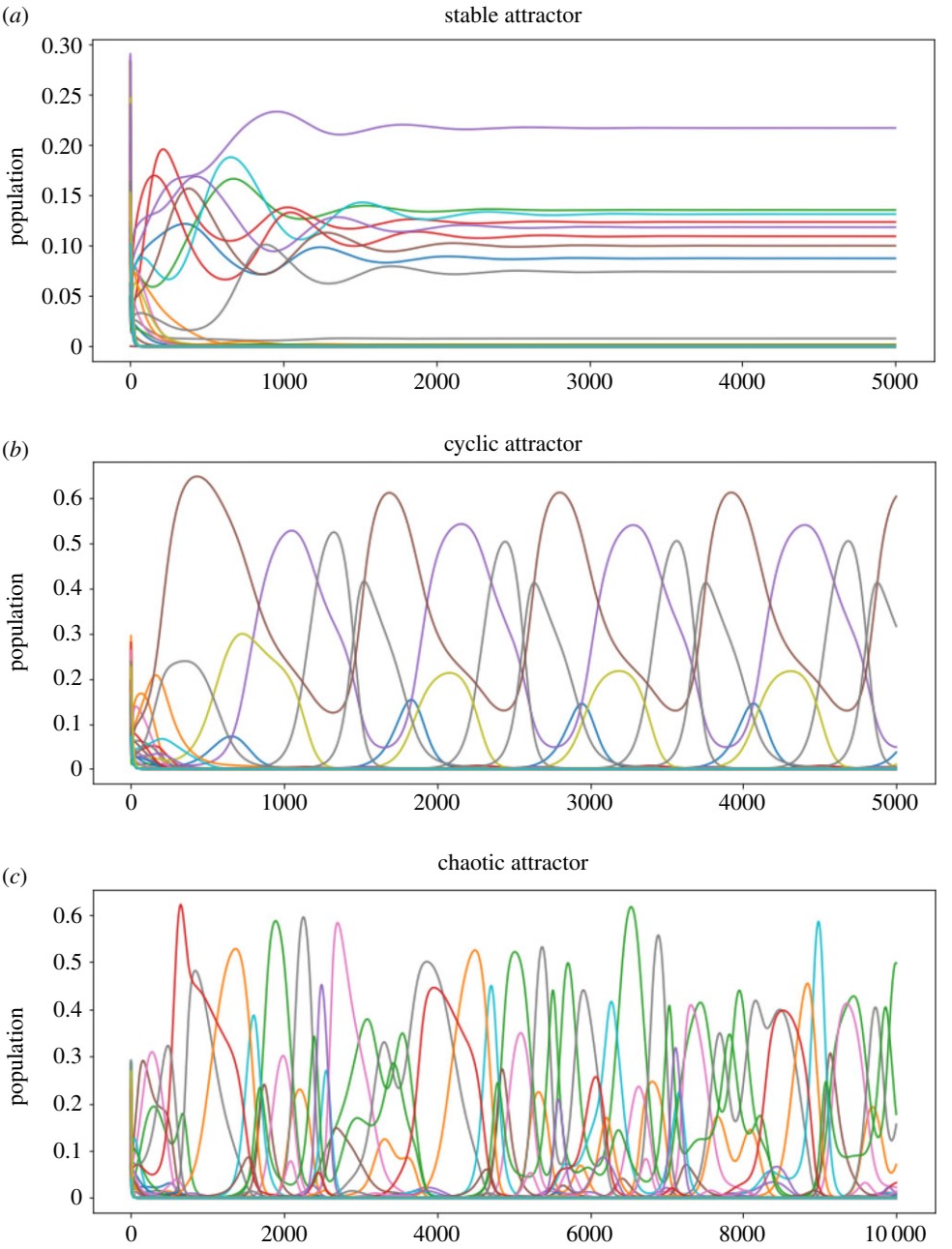

**Figure 2.** Our family of models generates time series of the population of each species. All of the three panels correspond to one simulation of an ecosystem initialized with 16 prey species and 12 predator species. Only the time series corresponding to prey populations are displayed. The time series can be classified in three qualitative types depending on their asymptotic behaviour: *stable*, *cyclic* and *chaotic*. In (*a*), the system reaches a stable attractor after a transient time. In (*b*), a periodic attractor, with an approximate period of 1000 days, is reached after the transient time. The system in (*c*) never reaches a stable nor a cyclic attractor, but a chaotic one.

each value of the competition parameter, 200 different initial conditions, predation and competition matrices were generated. The initial conditions were drawn from a uniform distribution between 1 and 2 mg l$^{-1}$, while the predation and competition matrices were drawn from the probability distributions described in §2.2.1. We used a Runge–Kutta solver (ode45) to simulate the model with each parameter set. A stabilizing run of 2000 days was executed to discard transient dynamics. Simulating for 5000 more days, we obtained a time series close to the attractor.

We determined the fraction of the 200 time series that were stable, cyclic or chaotic. For our multi-species models, we compared the performance of three different methods: visual inspection, estimation of the maximum Lyapunov exponent [23] and Gottwald and Melbourne's 0-1 test [24].

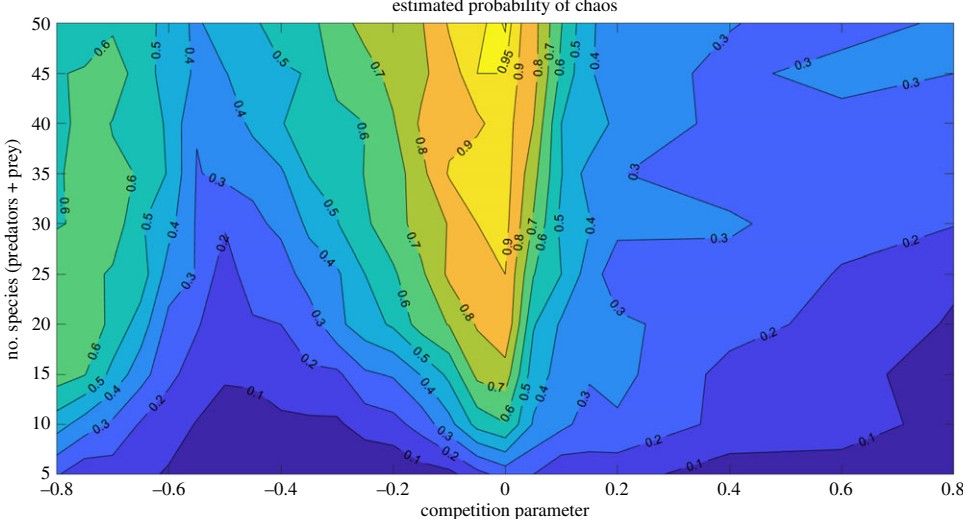

**Figure 3.** Contour map showing the probability of chaos for various competition parameters $\varepsilon$ (horizontal axis) and initial number of species of the simulation (vertical axis). The predators' species pool is fixed as two-thirds of the prey's species pool. Note that chaotic attractors appear more easily (i.e. for systems with less species) the closer is the competition to neutral (i.e. $\varepsilon = 0$).

We found the Gottwald–Melbourne test to be not only the most efficient but also the most reliable test for performing this classification. Describing in detail Gottwald and Melbourne's *0-1* test is beyond the scope of this paper, and it has already been done brilliantly in [24]. Nevertheless, a quick introduction to this test and how we applied it is given in electronic supplementary material, subsection A.1.

Additionally, two different measures of biodiversity were applied to each simulated ecosystem: the average number of non-extinct prey species and the average biomass grouped by trophic level. We considered a species to be extinct when their population density remained below a threshold of $0.01 \, \text{mg l}^{-1}$ after the stabilization run. We determined the relationship between the competition strength, the probability of each dynamical regime and the biodiversity.

The numerical experiment was repeated for species pools of different sizes, ranging from a total of 5 to 50 species. In our simulations, we kept a ratio of $2:3$ for the size of the species pool at the predator and the prey level.

In the spirit of reproducible research, we made available the code used to obtain our conclusions and generate our figures [25].

## 3. Results

From figure 3, we conclude that, in our model, the likelihood of chaotic dynamics reaches an optimum for near-neutral competition at the prey level. This result remains true for systems with a different number of species (see electronic supplementary material, figures A.3 and A.4). The likelihood of chaos also increases with the size of the food web. This effect should not be surprising: the more dimensions the phase space has, the easier is to fulfil the requirements of the complex geometry of a chaotic attractor [26]. Even in those higher-dimensional cases, there is still a clear maximum at near-neutral competition. The probability of chaos shows another local, lower maximum for weak competition coupling, while stable solutions are very rare (figure 4*a*). Possibly due to the weaker coupling we get less phase locking of the predator–prey cycles in this case.

Additionally, we found a clear correlation between the probability of chaos and the biodiversity. In all our cases, the diversity in systems with chaotic dynamics was highest (figure 4*b,c*) and the overall diversity peaked approximately at the near-neutral situation. Interestingly, also the cyclic solutions were clearly much more diverse than cases with stable dynamics (figure 4*b,c*). In fact, the difference in biodiversity of the situation with chaos and cycles was rather small (figure 4*c*). This conclusion remains true for food webs of different sizes (electronic supplementary material, figures A.5 and A.6). From figure 4*d*, we see that the prey biomass remains relatively stable for the whole range of competition parameters, with the exception of weak interspecific competition, where it reaches a maximum. The predator biomass grows almost linearly as the competition moves leftwards, from

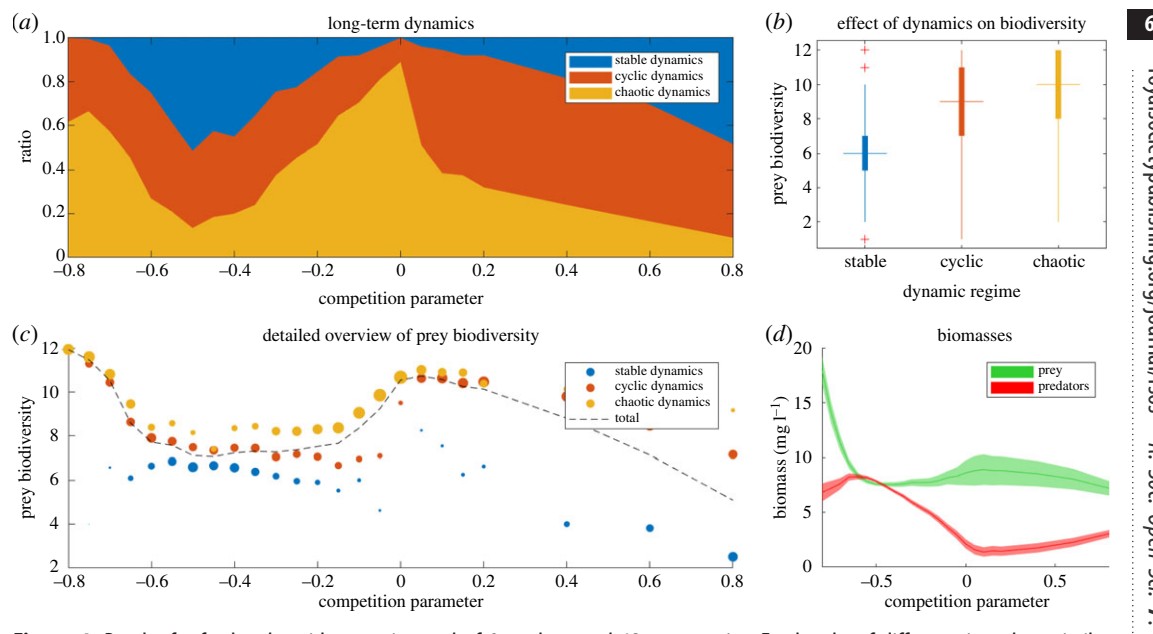

**Figure 4.** Results for food webs with a species pool of 8 predator and 12 prey species. Food webs of different sizes show similar results (see electronic supplementary material, section A.2). (*a*) Fraction of each dynamic regime as a function of the competition parameter. For each of the 25 parameter values, we simulated 200 ecosystems. (*b*) Box and whisker plot of the average number of non-extinct prey species grouped by asymptotic regime. (*c*). Average prey biodiversity as function of competition parameter $\varepsilon$. The dashed line shows the average number of non-extinct prey species grouped by competition parameter. The coloured circles represent the average prey biodiversity of the simulations, additionally grouped by dynamical regime (stable, cyclic and chaotic). The relative size of the circles represents the ratio of simulations that led to each kind of dynamics (d). Average biomasses grouped by trophic level versus competition parameter. The width represents standard deviation.

near-neutral to strong intraspecific, while the prey biomass remains constant. This also remains true for food webs of different sizes (see electronic supplementary material, figure A.7). We think this can be understood from the effect of niche complementarity which causes an increase in their total prey biomass [27]. Like in a two-species model, this increase in prey biomass results in an increase of predator biomass only (cf. [19]).

# 4. Discussion

We find that competition close to neutrality in simple food web models significantly increases the chances of complex dynamical behaviours (such as cycles and chaos), and also the biodiversity. These observations suggest that the hypothesis of non-equilibrium [13] and Hubbell's hypothesis of neutrality [7] are not completely independent. Our model shows another local maximum for the probability of chaos for weak competition coupling. We consider this a reasonable result, as predation is known to be the main driver of chaos in this kind of models [20]. Once again, this increase in the chances of complex dynamics is correlated with a higher biodiversity, but here the high biodiversity is also due to the low interspecific competition which obviously increases species coexistence.

With these results in hand, it may be tempting to conclude that chaos causes diversity. But this will be a premature conclusion. For instance, we cannot exclude that near-neutrality leads to a higher diversity and that this higher number of species makes chaotic dynamics more likely to occur. Teasing apart the exact pathway of causation is beyond the scope of this paper.

Our research question requires a fine control of the ecosystems under study and keeping a long-term track of their development in time. The experimental realization in a chemostat of a plankton ecosystem is very costly and time consuming even for a single run (cf. [17]). To study our research question experimentally, we would need many replicas and an experimental manipulation of the competition strengths. We think that such approach is unfeasible.

Our choice of the Rosenzweig–MacArthur model was based on the modeller's *mantra* of using the simplest possible model that shows the behaviour of interest. This model does not use Allee effects,

nor noise, nor species-specific carrying capacities, nor advanced parametrization techniques [28], and the functional form of each term has been chosen to account for satiation and saturation in the simplest possible ways. The presence of a random predation matrix $S$ may seem in contradiction with this pursuit of simplicity, but as we show in electronic supplementary material, figure A.8, it is indeed a requirement for the phenomenon to happen. It seems obvious that if all predators and prey were fully neutral (i.e. they all have the same parameters), there would not be chaos, as the system formally reduces to a two-species system, where chaos is ruled out by the Poincaré–Bendixson theorem [26].

Both the competition and predation parameter sets were drawn from probability distributions. The interactions in our system can be interpreted as a weighted network with a high connectivity. In nature, trophic networks tend to show modular structure with various clusters [29]. Our simplified model could be interpreted as representing one of those densely connected modules. Moreover, while in the present paper, our random parameters were drawn independently, the competition matrix can be chosen in a more advanced way (for instance, accounting for rock-paper-scissors competition). Studying the effect of different physiological scenarios (in the sense of [30], that is, constraints between the parameters) on the probabilities of chaos could be a continuation to this paper.

Our result seemed to be robust against changes in the number of species. However, the exact probabilities of cyclic or chaotic dynamics are of course dependent on the model details and on the values of all parameters [22]. For a system with such a high number of parameters, a systematic exploration of the parameter space is unfeasible. In the present work, we explored only the variation away from neutrality just by changing the competition strength and randomizing some of the parameters. We found no differences in the main qualitative results when our simulations were run under different sets of realistic parameters (*sensu* [22]). Two examples of such additional simulations, particularly using different values for the immigration parameter $f$, are available in electronic supplementary material, figure A.9.

Due to the large number of simulations made (there were 5000 simulated time series for each of the 10 different food web sizes analysed), we had to rely on automatic methods for detecting chaos. Automatic detection of chaos by numerical methods has fundamental limitations, especially for high-dimensional systems like ours. Most of them can be boiled down to the fact that, in general, numerical methods cannot distinguish robustly between long, complicated transients and genuine chaos. Our motivation to choose the Gottwald & Melbourne test [24] was threefold: it discriminates between stable, cyclic and chaotic, it scales easily to systems of higher dimensions, its computation is fast and it performs better than any other method we tried when compared with the visual inspection of the time series. Although we cannot exclude that we misinterpreted some of the generated time series due to long transients, we do not think this affected the overall patterns, as they were very robust in all our simulations.

Our results suggest a fundamentally new way in which near-neutrality may promote biodiversity. In addition to weakening the forces of competitive exclusion leading to long transients [9], our analyses reveal that near neutrality may boost the chances for more diverse chaotic and cyclic dynamics.

The results presented in this manuscript rely almost exclusively on simulations. Although they are beyond the scope of this manuscript, certain bridges with data from field ecology can be built. It is known that competition matrices can be estimated from field observations [31]. Provided some studies point in the direction of neutrality [8,10] and chaos [28] being an emergent phenomenon, we find it reasonable to expect near-neutral competition matrices to be common in real ecosystems. Assessing the interesting question of how frequent are near-neutral competition matrices in real ecosystems using data from field ecology represents a straightforward continuation of this manuscript.

Data accessibility. In the spirit of reproducible research, we published the code used to obtain our conclusions and generate our figures in Zenodo, an open access repository. It is available at the reference [25].

Authors' contributions. P.R.-S. designed and performed the numerical simulations, generated the figures and wrote the first drafts of the manuscript. E.v.N. conceived the idea that gave rise to this manuscript and supervised the simulations. Both E.v.N. and M.S. provided their expert knowledge of the literature in the field and contributed substantially to revisions. All authors gave final approval for publication.

Competing interests. We declare we have no competing interest.

Funding. This work was supported by funding from the European Union's *Horizon 2020* research and innovation programme for the *ITN CRITICS* under grant Agreement no. 643073.

Acknowledgements. The preliminary analysis of this model was performed using GRIND for Matlab (http://www.sparcs-center.org/grind). Additionally, we thank Tobias Oertel-Jäger, Sebastian Wieczorek, Peter Ashwin, Jeroen Lamb, Martin Rasmussen, Cristina Sargent, Jelle Lever, Moussa N'Dour, Iñaki Úcar, César Rodríguez and Sebastian Bathiany for their useful comments and suggestions.

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
