## [Reviewer comments · Royal Society Open Science]

Review History

RSOS-191532.R0 (Original submission)

Review form: Reviewer 1

Is the manuscript scientifically sound in its present form?

Yes

Are the interpretations and conclusions justified by the results?

No

Is the language acceptable?

Yes

Do you have any ethical concerns with this paper?

No

Have you any concerns about statistical analyses in this paper?

No

Recommendation?

Major revision is needed (please make suggestions in comments)

Comments to the Author(s)

All comments are in the attached file (Appendix A).

Review form: Reviewer 2**Is the manuscript scientifically sound in its present form?**

Yes

Are the interpretations and conclusions justified by the results?

Yes

Is the language acceptable?

Yes

Do you have any ethical concerns with this paper?

No

Have you any concerns about statistical analyses in this paper?

Yes

Recommendation?

Accept with minor revision (please list in comments)

Comments to the Author(s)

Thank you for this manuscript, it reads very nicely and is very interesting. I enjoyed going through the results and discussion sections to see what the authors have found in their analysis. There is one major comment I have for the authors, and other minor ones, that I believe can greatly improve the manuscript.

Major comment:

- Observations are not used in the analysis. Most/all of the conclusions are based on model simulations alone. I understand it can be quite difficult to find observed data on food webs (although c.f. Beninca et al 2008 which is cited in this paper), but measurements of food webs have shown that chaotic behavior is intrinsic to the dynamics of the system. The conclusions of the authors is based completely on the parameter values chosen to run the model simulations, which may be totally un-realistic model parameter values to use. The only way to really verify this is to compare the model simulations with observed data. Other studies have performed similar analysis which have fused model simulations with observations of food web dynamics (i.e. Massoud et al 2018 [1]), and they found that all calibrated parameter values that allowed the model simulations to fit the observed data exhibited chaotic dynamics. I recommend the authors at least cite this work and their findings and to include a brief discussion on how utilizing observed data can help with the conclusions of the study.

[1] Massoud, Elias C., Jef Huisman, Elisa Benincà, Michael C. Dietze, Willem Bouten, and Jasper A. Vrugt. "Probing the limits of predictability: data assimilation of chaotic dynamics in complex food webs." *Ecology letters* 21, no. 1 (2018): 93-103.

Minor Comments:

- The title of the paper should indicate that only model simulations are used. e.g. 'Neutral Competition boosts chaos in simulations of food webs'
- All the results are based on the default parameters listed in Table 1. But what if different values were used for these parameters? Will the final results be sensitive to this? Specifically, will

different parameter values produce a less likelihood/probability of chaotic dynamics in the model simulations? The authors should at least discuss this point in context of the final results.

- Section 4, Page 9 Line 3-4: The authors state "These observations suggest..." , yet the authors are referring to model simulations and not observations. I suggest the authors do include a brief discussion on observations, and sort out the vocabulary/terminology used in this section to properly discuss that the results in the paper are from model simulations and not observations.

- Section 4, Page 9 Line 7-8: The authors state "... it may be tempting to conclude that chaos causes diversity." I do not believe this is a valid conclusion, and one might make the opposite conclusion that 'diversity causes chaos'. I recommend the authors be careful when using this type of statement.

- Is there any way to include uncertainties with the final results? The authors should try to quantify their finding in the abstract/conclusions, e.g. up to 90% (+/- 5%) possibility of chaotic dynamics in the model simulations. A quantified conclusion with uncertainties can be very helpful for readers.

- Appendix Figures A2, A3, and A5 are not discuss in the main text. This makes it somewhat confusing to understand the role of these figures and what they add to the overall content of the manuscript.

Review form: Reviewer 3 (Angel Segura)

Is the manuscript scientifically sound in its present form?

Yes

Are the interpretations and conclusions justified by the results?

Yes

Is the language acceptable?

Yes

Do you have any ethical concerns with this paper?

No

Have you any concerns about statistical analyses in this paper?

Yes

Recommendation?

Accept with minor revision (please list in comments)

Comments to the Author(s)

GENERAL COMMENTS

The paper explore the effect of competition/neutrality on species diversity and population dynamics. Based on a simple multispecies predator/prey model, the authors explore the probability of chaotic dynamics in response to the average values of the competition matrix. The findings are novel and could inspire a new set of tests and experiments. I suggest to search for and include empirical examples (generated outside the group of investigators) in the line of chaos and biodiversity.

Some minor comments

Introduction Line 18-19- Hubells neutrality requires immigration for coexistence, as the final result is exclusion of all but one species.

Lines 22-23, also near neutrality does not lead to coexistence sensu-stricto, but it slow downs exclusion. Other mechanisms are required for coexistence.

Line 32- This sentence needs to be justified. Or at least it is not intuitive why should neutral communities are not chaotic. Please expand on this argument as it is central for the rest of the manuscript.

Methods

Please explain better the statistical test and how was it performed.

Discussion

Please expand on how to compare this theoretical results with experimental or natural data as to suggest some ways for testing this results..

Also, how common is to found near-neutral competition matrices in nature?

See for example

Fort, H. and Segura, A. 2017. Competition across diverse taxa: quantitative integration of theory and empirical research using global indices of competition. - *Oikos* doi: 10.1111/oik.04756

Or in some empirical estimations of competition matrix

Emergent neutrality drives phytoplankton species coexistence, 2011 PRSCB
<https://doi.org/10.1098/rspb.2010.2464>

Decision letter (RSOS-191532.R0)

12-Nov-2019

Dear Mr Rodríguez-Sánchez,

The editors assigned to your paper ("Neutral competition boosts chaos in food webs") have now received comments from reviewers. We would like you to revise your paper in accordance with the referee and Associate Editor suggestions which can be found below (not including confidential reports to the Editor). Please note this decision does not guarantee eventual acceptance.

Please submit a copy of your revised paper before 05-Dec-2019. Please note that the revision deadline will expire at 00.00am on this date. If we do not hear from you within this time then it will be assumed that the paper has been withdrawn. In exceptional circumstances, extensions may be possible if agreed with the Editorial Office in advance. We do not allow multiple rounds of revision so we urge you to make every effort to fully address all of the comments at this stage. If deemed necessary by the Editors, your manuscript will be sent back to one or more of the original reviewers for assessment. If the original reviewers are not available, we may invite new reviewers.

- Data accessibility

If you wish to submit your supporting data or code to Dryad (<http://datadryad.org/>), or modify your current submission to dryad, please use the following link:
<http://datadryad.org/submit?journalID=RSOS&manu=RSOS-191532>

- Competing interests

- Authors' contributions

- Acknowledgements

- Funding statement

Kind regards,
Lianne Parkhouse
Royal Society Open Science
openscience@royalsociety.org

on behalf of the Associate Editor, and Professor Kevin Padian (Subject Editor)
openscience@royalsociety.org

Associate Editor's comments:

Thank you for submitting your manuscript to Royal Society Open Science. Your manuscript has now returned from peer review, and three referees have provided both major and minor points that need addressing.

Both Referee #1 and Referee #2 commented on the interpretations and conclusions provided within your manuscript, and how they are not justified by the results of your study. Particularly, as specified by Referee #1, the discussion is too brief, and requires more in-depth interpretation. Referee #2 specified that the conclusions are based on simulations only, and that the authors should include a brief discussion on how utilizing observed data can help with the conclusions of the study.

Referee #3 commented on the clarity of how the GM test was conducted. Specifically, although the code appears in a repository, it is crucial to have more information in the paper on how the GM test was performed.

Please address the referees' comments appropriately, and ensure that you include a point-by-point response, detailing the changes made within your revision.

Comments to Author:

Reviewers' Comments to Author:

Reviewer: 1
Comments to the Author(s)

All comments are in the attached file.

Reviewer: 2
Comments to the Author(s)

Thank you for this manuscript, it reads very nicely and is very interesting. I enjoyed going through the results and discussion sections to see what the authors have found in their analysis. There is one major comment I have for the authors, and other minor ones, that I believe can greatly improve the manuscript.

Major comment:

- Observations are not used in the analysis. Most/all of the conclusions are based on model simulations alone. I understand it can be quite difficult to find observed data on food webs (although c.f. Beninca et al 2008 which is cited in this paper), but measurements of food webs have shown that chaotic behavior is intrinsic to the dynamics of the system. The conclusions of

the authors is based completely on the parameter values chosen to run the model simulations, which may be totally un-realistic model parameter values to use. The only way to really verify this is to compare the model simulations with observed data. Other studies have performed similar analysis which have fused model simulations with observations of food web dynamics (i.e. Massoud et al 2018 [1]), and they found that all calibrated parameter values that allowed the model simulations to fit the observed data exhibited chaotic dynamics. I recommend the authors at least cite this work and their findings and to include a brief discussion on how utilizing observed data can help with the conclusions of the study.

[1] Massoud, Elias C., Jef Huisman, Elisa Benincà, Michael C. Dietze, Willem Bouten, and Jasper A. Vrugt. "Probing the limits of predictability: data assimilation of chaotic dynamics in complex food webs." *Ecology letters* 21, no. 1 (2018): 93-103.

Minor Comments:

- The title of the paper should indicate that only model simulations are used. e.g. 'Neutral Competition boosts chaos in simulations of food webs'
- All the results are based on the default parameters listed in Table 1. But what if different values were used for these parameters? Will the final results be sensitive to this? Specifically, will different parameter values produce a less likelihood/probability of chaotic dynamics in the model simulations? The authors should at least discuss this point in context of the final results.
- Section 4, Page 9 Line 3-4: The authors state "These observations suggest..." , yet the authors are referring to model simulations and not observations. I suggest the authors do include a brief discussion on observations, and sort out the vocabulary/terminology used in this section to properly discuss that the results in the paper are from model simulations and not observations.
- Section 4, Page 9 Line 7-8: The authors state "... it may be tempting to conclude that chaos causes diversity." I do not believe this is a valid conclusion, and one might make the opposite conclusion that 'diversity causes chaos'. I recommend the authors be careful when using this type of statement.
- Is there any way to include uncertainties with the final results? The authors should try to quantify their finding in the abstract/conclusions, e.g. up to 90% (+/- 5%) possibility of chaotic dynamics in the model simulations. A quantified conclusion with uncertainties can be very helpful for readers.
- Appendix Figures A2, A3, and A5 are not discuss in the main text. This makes it somewhat confusing to understand the role of these figures and what they add to the overall content of the manuscript.

Reviewer: 3

Comments to the Author(s)

GENERAL COMMENTS

The paper explore the effect of competition/neutrality on species diversity and population dynamics. Based on a simple multispecies predator/prey model, the authors explore the probability of chaotic dynamics in response to the average values of the competition matrix. The findings are novel and could inspire a new set of tests and experiments. I suggest to search for and include empirical examples (generated outside the group of investigators) in the line of chaos and biodiversity.

Some minor comments

Introduction Line 18-19- Hubells neutrality requires immigration for coexistence, as the final result is exclusion of all but one species.

Lines 22-23, also near neutrality does not lead to coexistence sensu-stricto, but it slow downs exclusion. Other mechanisms are required for coexistence.

Line 32- This sentence needs to be justified. Or at least it is not intuitive why should neutral communities are not chaotic. Please expand on this argument as it is central for the rest of the manuscript.

Methods

Please explain better the statistical test and how was it performed.

Discussion

Please expand on how to compare this theoretical results with experimental or natural data as to suggest some ways for testing this results..

Also, how common is to found near-neutral competition matrices in nature?

See for example

Fort, H. and Segura, A. 2017. Competition across diverse taxa: quantitative integration of theory and empirical research using global indices of competition. – *Oikos* doi: 10.1111/oik.04756

Or in some empirical estimations of competition matrix

Emergent neutrality drives phytoplankton species coexistence, 2011 PRSCB

<https://doi.org/10.1098/rspb.2010>.

Author's Response to Decision Letter for (RSOS-191532.R0)

See Appendix B.

RSOS-191532.R1 (Revision)

Review form: Reviewer 1

Is the manuscript scientifically sound in its present form?

No

Are the interpretations and conclusions justified by the results?

Yes

Is the language acceptable?

Yes

Do you have any ethical concerns with this paper?

No

Have you any concerns about statistical analyses in this paper?

No

Recommendation?

Major revision is needed (please make suggestions in comments)

Comments to the Author(s)

All comments are in the attached file (Appendix C).

Review form: Reviewer 2

Is the manuscript scientifically sound in its present form?

Yes

Are the interpretations and conclusions justified by the results?

Yes

Is the language acceptable?

Yes

Do you have any ethical concerns with this paper?

No

Have you any concerns about statistical analyses in this paper?

No

Recommendation?

Accept as is

Comments to the Author(s)

Thank you and good job on the revisions.

Decision letter (RSOS-191532.R1)

06-Apr-2020

Dear Mr Rodríguez-Sánchez:

Manuscript ID RSOS-191532.R1 entitled "Neutral competition boosts cycles and chaos in simulated food webs" which you submitted to Royal Society Open Science, has been reviewed. The comments of the reviewer(s) are included at the bottom of this letter.

Please submit a copy of your revised paper before 29-Apr-2020. Please note that the revision deadline will expire at 00.00am on this date. If we do not hear from you within this time then it will be assumed that the paper has been withdrawn. In exceptional circumstances, extensions may be possible if agreed with the Editorial Office in advance. We do not allow multiple rounds of revision so we urge you to make every effort to fully address all of the comments at this stage. If deemed necessary by the Editors, your manuscript will be sent back to one or more of the original reviewers for assessment. If the original reviewers are not available we may invite new reviewers.

When submitting your revised manuscript, you must respond to the comments made by the referees and upload a file "Response to Referees" in "Section 6 - File Upload". Please use this to document how you have responded to the comments, and the adjustments you have made. In

order to expedite the processing of the revised manuscript, please be as specific as possible in your response.

- Ethics statement

- Data accessibility

- Competing interests

- Authors' contributions

- Acknowledgements

- Funding statement

Kind regards,
Lianne Parkhouse
Editorial Coordinator
Royal Society Open Science
openscience@royalsociety.org

on behalf of the Associate Editor, and Professor Kevin Padian (Subject Editor)
openscience@royalsociety.org

Associate Editor Comments to Author:

Thank you for the revised paper - while the reviewers observe that you have made efforts to tackle their earlier queries, one of the referees comments that two of their major concerns remain to be satisfactorily answered. Generally, we don't allow multiple rounds of revision, but exceptions can be made where it appears the authors have made efforts to meet the reviewers' concerns, as seems to be broadly the case here. We're going to give you the benefit of the doubt on this occasion, so you can have a final go at revising the paper - but please bear in mind that if the paper doesn't 'get over the line' this time, we won't be able to grant further exceptions. If you've any queries, please contact the editorial office at the appropriate email address, and they'll be glad to assist. Good luck!

Reviewer comments to Author:

Reviewer: 1
Comments to the Author(s)

All comments are in the attached file.

Reviewer: 2
Comments to the Author(s)

Thank you and good job on the revisions.

Author's Response to Decision Letter for (RSOS-191532.R1)

See Appendix D.

RSOS-191532.R2 (Revision)

Review form: Reviewer 1

Is the manuscript scientifically sound in its present form?

Yes

Are the interpretations and conclusions justified by the results?

Yes

Is the language acceptable?

Yes

Do you have any ethical concerns with this paper?

No

Have you any concerns about statistical analyses in this paper?

No

Recommendation?

Accept as is

Comments to the Author(s)

Thank you for addressing both major as well as the minor comments I had in a satisfactory matter. It is now clear that the predator feeding preferences are needed to observe the phenomenon discussed in the manuscript.

I spotted one tiny spelling error: page 15 of 45 l. 14: "... were chaos is ruled out ..." should be "... where chaos is ruled out ...".

Decision letter (RSOS-191532.R2)

18-May-2020

Dear Mr Rodríguez-Sánchez,

It is a pleasure to accept your manuscript entitled "Neutral competition boosts cycles and chaos in simulated food webs" in its current form for publication in Royal Society Open Science. The comments of the reviewer(s) who reviewed your manuscript are included at the foot of this letter.

You can expect to receive a proof of your article in the near future. Please contact the editorial office (openscience_proofs@royalsociety.org) and the production office (openscience@royalsociety.org) to let us know if you are likely to be away from e-mail contact -- if

you are going to be away, please nominate a co-author (if available) to manage the proofing process, and ensure they are copied into your email to the journal.

Kind regards,

Anita Kristiansen
Editorial Coordinator

on behalf of Kevin Padian (Subject Editor)
openscience@royalsociety.org

Associate Editor Comments to Author:

Comments to the Author:

The paper is now recommended for publication: note that the reviewer has identified a minor spelling error that can be corrected during proofing. Congratulations on your manuscript and thank you for the submission to the journal.

Reviewer comments to Author:

Reviewer: 1

Comments to the Author(s)

Thank you for addressing both major as well as the minor comments I had in a satisfactory matter. It is now clear that the predator feeding preferences are needed to observe the phenomenon discussed in the manuscript.

I spotted one tiny spelling error: page 15 of 45 l. 14: "... were chaos is ruled out ..." should be "... where chaos is ruled out ...".

Appendix A

This manuscript studies the link between prey competition, type of dynamics (stable, cyclic, or chaotic), and species richness, using a multi-species model in which the prey competition strength can be varied from mainly intra-specific to mainly inter-specific. When competition is neutral, i.e., in the middle between these two extremes, the model is predisposed towards behaving chaotically, and tends to be rich in species.

Overall, I enjoyed reading the manuscript. It is very interesting to learn how the relatively simple basis of the model used by the Authors can lead to the observed variations in type of dynamics and species richness, and the Authors explain these two different aspects well. In addition, I highly appreciate the effort of the Authors to provide the code necessary to reproduce the results and play around with the model yourself.

However, I also have some concerns which should be addressed and would increase the manuscript's clarity and scientific quality.

Major points

1. One of the main findings the Authors claim to find is that chaotic dynamics correlate with a higher biodiversity. This is mentioned in several locations throughout the manuscript, but I am skeptical whether the data presented really supports this conclusion. The difference in “prey biodiversity” between cyclic and chaotic dynamics as presented in Figure 4b (as well as for other amounts of initial species in the appendix, Figure A.4) does not seem significant. However, each of these figures does show a strong difference between either stable dynamics, and cyclic or chaotic dynamics. Therefore, it seems to me that it would be more accurate to state that both cyclic and chaotic dynamics correlate with a higher biodiversity when compared to stable dynamics. Currently, lines 5-10 on page 9 read particularly confusing, as the Authors first state that also non-chaotic periodic dynamics are correlated with a higher biodiversity, and then state that it would be tempting to claim that chaos causes diversity.
2. My main concern is that the manuscript lacks depth regarding the interpretation of the results obtained. A more elaborate discussion would strongly improve the quality of the manuscript.

For example, in the opening of the Discussion (lines 2 – 11 on page 9), the Authors basically state that they cannot conclude whether purely chaos leads to more diversity, or that it is the near-neutrality which leads to more diversity, which in turn increases the likelihood of chaotic dynamics. I'm not proposing that this puzzle should be solved in this manuscript, but a few more words here would be informative and relevant. Using Figures 1, 2a and 2c as a guide, it does seem that there is a correlation between chaos and species diversity irrespectively of the type of competition. As mentioned in the Results section (on lines 29 – 31, page 7), there is another local maximum at the intra-specific dominating case for which the proportion of chaotic dynamics is also high.

In the following paragraphs (page 9, lines 12 – 30) the Authors mention that more complex interactions or other model adjustments might affect the results obtained.

Also here, a few more elaborating words or perhaps predictions regarding potential influences would be very informative.

3. The Authors mention that the parameters used for obtaining the results (Section 2.2) are based on values from Reference 21. In that publication, many of the parameters are varied within certain ranges, while the present manuscript only uses one sample out of these intervals. Some motivation for why these specific values were chosen, and whether or not this choice is expected to have a certain influence on the results obtained is lacking.

In particular, I wonder whether the specific value of the immigration rate $f = 10^{-5}$ has any significance (why was this value chosen?), and how strongly it would influence the results obtained. It is easy to imagine how this parameter might have a crucial influence under near-neutrality. In this case, the fitness differences between the prey species might be very small, and hence, some would be outcompeted by others, but only on a long timescale. The immigration rate, even though it is small, might make this effect unobservable and be sufficient in itself for the species to coexist.

4. The Authors define the lower border of the range within which ϵ is varied at $\epsilon = -0.8$, to prevent the elements of the competition matrix to become negative. However, as the width of the distribution from which the elements of the competition matrix A are sampled is fixed at $w = 0.05$, wouldn't that mean that, in theory, the minimal value for ϵ would be $\epsilon = -0.975$? If this is indeed the case, why did the Authors settle on $\epsilon = -0.8$ as the minimal value?
5. In any case, given the minimum of $\epsilon = -0.8$, the Authors claim that, as this is a reduction from the neutral case by 0.8, the upper limit is determined symmetrically by setting the upper limit to $\epsilon = 0.8$. However, as the elements of the competition matrix A affect the state variables in a multiplicative way, I wonder whether it might make more sense to approach this geometrically: for $\epsilon = -0.8$ the elements of A are centered around 0.2, which is a 5-fold reduction from the neutral case of 1, whereas $\epsilon = 0.8$ corresponds only to an increase by a factor of 1.8.

It would be helpful to see at least some confirmation that the current window of $\epsilon \in [-0.8, 0.8]$ captures all interesting behaviour possible in the model, or otherwise some ecological legitimation for the upper bound of $\epsilon = 0.8$ should be given.

6. The Authors state that the elements of the predation matrix S are sampled randomly between 0 and 1. For this large range, even in the near-neutrality case, strong differences may arise between the prey species, which makes it hard to imagine how exactly it is influencing the system as a whole. For example, this mechanism alone can lead to cycles in biomass between certain predators and their preferential prey.

Why did the Authors settle for this potentially complicated mechanism? Might it not be simpler to use a density-dependent functional response (e.g. Holling-type III) between a predator and multiple prey species, in order to make the results easier to interpret?

Minor points

7. While the language of the manuscript is almost everywhere understandable, in a few cases some confusion may arise:
 - pg. 6, l. 6: “The diagonal terms are by definition *equal to 1*.” Without the *equal to* one might expect a Definition 1 to appear somewhere.
 - pg. 6, l. 45: “assure” should be “ensure”.
 - pg. 9, l. 37” “escalates”, should maybe be “scales”?

In addition, the text should be streamlined with respect to the term used for either the consumers or predators, using both may be confusing.

8. The Authors talk about both diversity and measure this using the species number. I wonder whether the results are strongly affected when a diversity index (e.g. Shannon index) is used instead, which would be less sensitive to the particular value of the “extinction threshold” the Authors used.
9. In the current model formulation, the prey species more or less share the carrying capacity K in the near-neutral case, whereas in the intra-specific dominating case each prey species more or less has its own carrying capacity K . Might this property alone not explain the sharp increase in prey biomass for the intra-specific dominating case, shown in figure 4d?
10. Why did the Authors use a fixed ratio of 2 : 3 between initial predator and prey species numbers? It would be relevant to show that this does not have a strong influence on the results obtained, or otherwise ecologically motivate the choice for this particular ratio.
11. Figure 2: I understand that this figure is there to prove that the different dynamical types are indeed possible in the model, and to show readers what they look like. However, I think that a bit more information on what is actually shown may necessary. In particular, for which parameter combination were these figures produced, and for how many (initial) species? Which lines are producers, and which lines are consumers?
12. pg. 7, l. 2: More information on how the “random initial conditions” were obtained would be interesting. In addition, it should be mentioned whether the elements of the competition matrix A and predation matrix S were also randomly sampled for each of the 200 runs per combination of species numbers and competition parameter ϵ , or whether they remained fixed.
13. pg. 7, l. 31-32: Here more information on why weaker coupling would lead to less phase coupling, and why less phase coupling would lead either to more chaotic solutions or less stable solutions is required as well, to understand what exactly the Authors mean.
14. pg 7, l 43-48: I am not able to follow the Authors’ reasoning clearly in this part of the results, with regards to niche complementarity leading to an increase in “effective”

prey biomass leading to an increase in predator biomass. If this is an important result it should be explained more clearly.

15. Figure 3: Please clarify whether the “Number of species” on the y -axis of Figure 3 is the number of species for which the simulation was initiated, or whether it is the number of coexisting species present after equilibration to the attractor. In addition, what is meant by “smaller systems” as mentioned in the caption?
16. Figure 4: Please clarify what n is here (mentioned in the caption).

Appendix B

Editor's comments

Dear Mr Rodríguez-Sánchez,

The editors assigned to your paper ("Neutral competition boosts chaos in food webs") have now received comments from reviewers. We would like you to revise your paper in accordance with the referee and Associate Editor suggestions which can be found below (not including confidential reports to the Editor). Please note this decision does not guarantee eventual acceptance.

Please submit a copy of your revised paper before 05-Dec-2019. Please note that the revision deadline will expire at 00.00am on this date. If we do not hear from you within this time then it will be assumed that the paper has been withdrawn. In exceptional circumstances, extensions may be possible if agreed with the Editorial Office in advance. We do not allow multiple rounds of revision so we urge you to make every effort to fully address all of the comments at this stage. If deemed necessary by the Editors, your manuscript will be sent back to one or more of the original reviewers for assessment. If the original reviewers are not available, we may invite new reviewers.

- Data accessibility

It is a condition of publication that all supporting data are made available either as supplementary information or preferably in a suitable permanent repository. The data accessibility section should state where the article's supporting data can be accessed. This section should also include details, where possible of where to access other relevant research materials such as statistical tools, protocols, software etc can be accessed. If the data have been deposited in an external repository this section should list the database, accession number

and link to the DOI for all data from the article that have been made publicly available. Data sets that have been deposited in an external repository and have a DOI should also be appropriately cited in the manuscript and included in the reference list.

<http://datadryad.org/submit?journalID=RSOS&manu=RSOS-191532>

- **Competing interests**

- **Authors' contributions**

- **Acknowledgements**

- **Funding statement**

Kind regards,

Lianne Parkhouse
Royal Society Open Science
openscience@royalsociety.org

on behalf of the Associate Editor, and Professor Kevin Padian (Subject Editor)
openscience@royalsociety.org

Associate Editor's comments

Thank you for submitting your manuscript to Royal Society Open Science. Your manuscript has now returned from peer review, and three referees have provided both major and minor points that need addressing.

Both Referee #1 and Referee #2 commented on the interpretations and conclusions provided within your manuscript, and how they are not justified by the results of your study. Particularly, as specified by Referee #1, the discussion is too brief, and requires more in-depth interpretation. Referee #2 specified that the conclusions are based on simulations only, and that the authors should include a brief discussion on how utilizing observed data can help with the conclusions of the study.

Referee #3 commented on the clarity of how the GM test was conducted. Specifically, although the code appears in a repository, it is crucial to have more information in the paper on how the GM test was performed.

Please address the referees' comments appropriately, and ensure that you include a point-by-point response, detailing the changes made within your revision.

Dear editors and reviewers.

Attached you'll find an updated manuscript of our previous submission ('Neutral competition boosts chaos in food webs', RSOS-191532.R1). Please note that based on the reviewers' suggestions we changed the title of the manuscript.

We would like to thank you and the reviewers for your useful comments. All of the suggestions have been carefully considered, and most of them have been implemented in the updated version of the manuscript. The present document shows, in green, our responses to the reviewers.

In order to facilitate the review process, we provide a file highlighting the differences between the new and the previous version. Additionally, we added the sections of Data accessibility, Competing interests, Authors' contributions and Funding statement.

The authors.

Comments to Author:

Reviewers' Comments to Author:

Reviewer: 1

Comments to the Author(s) (originally in attached file)

This manuscript studies the link between prey competition, type of dynamics (stable, cyclic, or chaotic), and species richness, using a multi-species model in which the prey competition strength can be varied from mainly intra-specific to mainly inter-specific. When competition is neutral, i.e., in the middle between these two extremes, the model is predisposed towards behaving chaotically, and tends to be rich in species.

Overall, I enjoyed reading the manuscript. It is very interesting to learn how the relatively simple basis of the model used by the Authors can lead to the observed variations in type of dynamics and species richness, and the Authors explain these two different aspects well. In addition, I highly appreciate the effort of the Authors to provide the code necessary to reproduce the results and play around with the model yourself.

However, I also have some concerns which should be addressed and would increase the manuscript's clarity and scientific quality.

Major points

1. One of the main findings the Authors claim to find is that chaotic dynamics correlate with a higher biodiversity. This is mentioned in several locations throughout the manuscript, but I am skeptical whether the data presented really supports this conclusion. The difference in "prey biodiversity" between cyclic and chaotic dynamics as presented in Figure 4b (as well as for other amounts of initial species in the appendix, Figure A.4) does not seem significant. However, each of these figures does show a strong difference between either stable dynamics, and cyclic or chaotic dynamics. Therefore, it seems to me that it would be more accurate to state that both cyclic and chaotic dynamics correlate with a higher biodiversity when compared to stable dynamics. Currently, lines 5-10 on page 9 read particularly confusing, as the Authors first state that also non-chaotic periodic dynamics are correlated with a higher biodiversity, and then state that it would be tempting to claim that chaos causes diversity.

We fully agree. Based on literature our starting point was that mainly chaos promotes biodiversity, but our own data shows that the regular cycles that we see have almost the same effect. This makes intuitively much sense as in both ways species can avoid competitive exclusion. The new version is more explicit about the fact that both cyclic and chaotic dynamics correlate with a higher biodiversity, to the point that we even modified the title.

2. My main concern is that the manuscript lacks depth regarding the interpretation of the results obtained. A more elaborate discussion would strongly improve the quality of the manuscript.

For example, in the opening of the Discussion (lines 2 – 11 on page 9), the Authors basically state that they cannot conclude whether purely chaos leads to more diversity, or that it is the near-neutrality which leads to more diversity, which in turn increases the likelihood of chaotic dynamics. I'm not proposing that this puzzle should be solved in this manuscript, but a few more words here would be informative and relevant. Using Figures 1, 2a and 2c as a guide, it does

seem that there is a correlation between chaos and species diversity irrespectively of the type of competition. As mentioned in the Results section (on lines 29 – 31, page 7), there is another local maximum at the intra-specific dominating case for which the proportion of chaotic dynamics is also high.

In the following paragraphs (page 9, lines 12 – 30) the Authors mention that more complex interactions or other model adjustments might affect the results obtained.

Also here, a few more elaborating words or perhaps predictions regarding potential influences would be very informative.

The Discussion section has been expanded in this revised version. It now includes an explanation of our choice of a mathematical model to solve our research question, possible links with field data and an explicit mention to the local maximum corresponding to the intra-specific dominating case.

Regarding the other two suggestions, after careful consideration we decided to not include them in the Discussion section. These are our reasons:

The causation pathway is impossible to derive from a numerical experiment, and we don't see how using figures 1 and 2 (added exclusively for pedagogical reasons) can help. We think it is more honest to openly address the fact that while we can prove correlation, we cannot say much about causation.

Our statement about different models yielding, potentially, different results, is a mere recognition of the general limitations of any mathematical model. In this paper we analyze only one family of models, and making reliable predictions about other families of models will require simulating them.

3. The Authors mention that the parameters used for obtaining the results (Section 2.2) are based on values from Reference 21. In that publication, many of the parameters are varied within certain ranges, while the present manuscript only uses one sample out of these intervals. Some motivation for why these specific values were chosen, and whether or not this choice is expected to have a certain influence on the results obtained is lacking.

In Dakos et al. different parameters were drawn at random from ranges that represent real phytoplankton species. That meant that the species differed rather much in their growth rates etc. Here we thought it is better to minimize the random assigned parameters to represent the neutral competition. It is especially relevant to keep the carrying capacity fixed otherwise one species will outcompete the others. (this was also done in Dakos). We changed the text accordingly.

In particular, I wonder whether the specific value of the immigration rate $f = 10^{-5}$ has any significance (why was this value chosen?), and how strongly it would influence the results obtained. It is easy to imagine how this parameter might have a crucial influence under near-neutrality. In this case, the fitness differences between the prey species might be very small, and hence, some would be outcompeted by others, but only on a long timescale. The immigration rate, even though it is small, might make this effect unobservable and be sufficient in itself for the species to coexist.

Without this immigration factor we get frequently heteroclinic cycles (see for instance Van Nes 2004 <https://doi.org/10.1086/422204>). We think these heteroclinic cycles are a model artifact as it involves cycles where some species get unrealistically low values (say 10^{-4} or even lower). Indeed the side effect could be that we keep some rare species alive. However as we use a threshold in the determination of diversity, these species are not counted in the diversity. This is explicitly stated in the "Model description" subsection.

4. The Authors define the lower border of the range within which e is varied at $e = -0.8$, to prevent the elements of the competition matrix to become negative. However, as the width of the distribution from which the elements of the competition matrix A are sampled is fixed at $w = 0.05$, wouldn't that mean that, in theory, the minimal value for e would be $e = -0.975$? If this is indeed the case, why did the Authors settle on $e = -0.8$ as the minimal value?

We are afraid there is some misunderstanding here. The minimal value for e is indeed -0.8 . The minimal value in the competition matrix elements is thus $1 - 0.8 - 0.05 = 0.15$. We did not mean to test very low values of the competition matrix, as it is obvious that without competition all species can coexist as there is then only intraspecific competition.

We clarified that our choice of a minimal e of -0.8 with the sentence "The lower value was chosen to assure that the non-diagonal competition matrix elements were positive and non-negligible to exclude facilitation or non-competing species".

5. In any case, given the minimum of $e = -0.8$, the Authors claim that, as this is a reduction from the neutral case by 0.8 , the upper limit is determined symmetrically by setting the upper limit to $e = 0.8$. However, as the elements of the competition matrix A affect the state variables in a multiplicative way, I wonder whether it might make more sense to approach this geometrically: for $e = -0.8$ the elements of A are centered around 0.2 , which is a 5-fold reduction from the neutral case of 1 , whereas $e = 0.8$ corresponds only to an increase by a factor of 1.8 .

It would be helpful to see at least some confirmation that the current window of $e \in [-0.8, 0.8]$ captures all interesting behaviour possible in the model, or otherwise some ecological legitimation for the upper bound of $e = 0.8$ should be given.

While there is an obvious lower limit for e ($e = -1$, the one that switches competition into cooperation), there is not such an obvious limit for the upper (as organisms can compete more and more aggressively). In this sense, any upper limit could be questioned.

We think the range that we tested is a bit higher than realistic for algal species, which was reported by Dakos, et.al. (2009) to be in the range of $0.5-1.5$ ($e = -0.5$ to 0.5). We did this to ensure we captured all interesting behaviour..

6. The Authors state that the elements of the predation matrix S are sampled randomly between 0 and 1 . For this large range, even in the near-neutrality case, strong differences may arise between the prey species, which makes it hard to imagine how exactly it is influencing the system as a whole. For example, this mechanism alone can lead to cycles in biomass between certain predators and their preferential prey.

Why did the Authors settle for this potentially complicated mechanism? Might it not be simpler to use a density-dependent functional response (e.g. Holling-type III) between a predator and multiple prey species, in order to make the results easier to interpret?

Following Dakos et al. 2009, we used the selectivity matrix S to model food preference of predators in a simple way. Of course there are other ways of modelling this, but it is beyond the scope of our paper to study the effects of other functional responses. We are not sure why the reviewer thinks the Holling-type III functional response is easier to interpret than the Holling type II with preferences.

Minor points

7. While the language of the manuscript is almost everywhere understandable, in a few cases some confusion may arise:

- pg. 6, l. 6: “The diagonal terms are by definition equal to 1.” Without the equal to one might expect a Definition 1 to appear somewhere.
- pg. 6, l. 45: “assure” should be “ensure”.
- pg. 9, l. 37 “escalates”, should maybe be “scales”?

In addition, the text should be streamlined with respect to the term used for either the consumers or predators, using both may be confusing.

All the above-mentioned corrections have been accepted and applied. The text has been scanned for the terms producer and consumer, and substituted by prey and predator.

8. The Authors talk about both diversity and measure this using the species number. I wonder whether the results are strongly affected when a diversity index (e.g. Shannon index) is used instead, which would be less sensitive to the particular value of the “extinction threshold” the Authors used.

Indeed there are many indicators of diversity that also include the evenness like the Shannon index. We prefer the number of species as we think this is more intuitive. Moreover we think it is better not to focus on the rare species as they are strongly dependent on the small import term (see above point 3).

9. In the current model formulation, the prey species more or less share the carrying capacity K in the near-neutral case, whereas in the intra-specific dominating case each prey species more or less has its own carrying capacity K . Might this property alone not explain the sharp increase in prey biomass for the intra-specific dominating case, shown in figure 4d?

That interpretation is correct, but this is only true for the prey biomass. We added some explicit explanation in the discussion.

10. Why did the Authors use a fixed ratio of 2 : 3 between initial predator and prey species numbers? It would be relevant to show that this does not have a strong influence on the results obtained, or otherwise ecologically motivate the choice for this particular ratio.

This ratio reflects the ratio in the species pools and not in the resulting model systems. Therefore we think this ratio is not critical and we choose an arbitrary value. Our choice was motivated for having a reasonable ratio to work with. Exploring all possible ratios is as unfeasible as exploring the whole parameter space.

11. Figure 2: I understand that this figure is there to prove that the different dynamical types are indeed possible in the model, and to show readers what they look like. However, I think that a bit more information on what is actually shown may be necessary. In particular, for which parameter combination were these figures produced, and for how many (initial) species? Which lines are producers, and which lines are consumers?

We extended the caption as suggested, with the exception of explicitly citing the parameter combination. The amount of parameters needed to generate each figure is too big to be printed (it roughly amounts to square the number of involved prey species), and this figure has only pedagogical purposes. Figures like this can be easily produced with the code we provided.

12. pg. 7, l. 2: More information on how the “random initial conditions” were obtained would be interesting. In addition, it should be mentioned whether the elements of the competition matrix A and predation matrix S were also randomly sampled for each of the 200 runs per combination of species numbers and competition parameter, or whether they remained fixed.

The random distribution corresponding to the initial conditions is now specified.

Both matrices are randomly sampled in each individual timeseries. This was explicitly mentioned in subsection 2.3 Numerical experiments. We rephrased this more explicitly.

13. pg. 7, l. 31-32: Here more information on why weaker coupling would lead to less phase coupling, and why less phase coupling would lead either to more chaotic solutions or less stable solutions is required as well, to understand what exactly the Authors mean.

If there is coupling of the phases of all predator-prey cycles there obviously is one cycle of all the species that are present. If there are different phases in the predator-prey cycles more complex cycles like chaos are possible. Phase locking also leads to less stable solutions as it always leads to a limit cycle.

14. pg 7, l 43-48: I am not able to follow the Authors’ reasoning clearly in this part of the results, with regards to niche complementarity leading to an increase in “effective” prey biomass leading to an increase in predator biomass. If this is an important result it should be explained more clearly.

We did not mean to say that the effective prey biomass increases with niche complementarity, but just that effectively the total prey biomass increases. This is due to niche complementarity leading to higher total biomasses (in the extreme case of no competition, all carrying capacities add up). As this is not a main result of this study we added a reference to Schnitzer et al. where this effect is explained in more detail.

15. Figure 3: Please clarify whether the “Number of species” on the y-axis of Figure 3 is the number of species for which the simulation was initiated, or whether it is the number of coexisting species present after equilibration to the attractor. In addition, what is meant by “smaller systems” as mentioned in the caption?

We agree that this can be confusing we now call the initial number of species the “species pool” throughout the manuscript.

16. Figure 4: Please clarify what n is here (mentioned in the caption).

n referred to the number of simulated ecosystems per competition parameter. We agree that this was not clear, and updated this caption with a more transparent explanation.

Reviewer: 2

Comments to the Author(s)

Thank you for this manuscript, it reads very nicely and is very interesting. I enjoyed going through the results and discussion sections to see what the authors have found in their analysis. There is one major comment I have for the authors, and other minor ones, that I believe can greatly improve the manuscript.

Major comment:

- Observations are not used in the analysis. Most/all of the conclusions are based on model simulations alone. I understand it can be quite difficult to find observed data on food webs (although c.f. Benincà et al 2008 which is cited in this paper), but measurements of food webs have shown that chaotic behavior is intrinsic to the dynamics of the system. The conclusions of the authors is based completely on the parameter values chosen to run the model simulations, which may be totally un-realistic model parameter values to use. The only way to really verify this is to compare the model simulations with observed data. Other studies have performed similar analysis which have fused model simulations with observations of food web dynamics (i.e. Massoud et al 2018 [1]), and they found that all calibrated parameter values that allowed the model simulations to fit the observed data exhibited chaotic dynamics. I recommend the authors at least cite this work and their findings and to include a brief discussion on how utilizing observed data can help with the conclusions of the study.

[1] Massoud, Elias C., Jef Huisman, Elisa Benincà, Michael C. Dietze, Willem Bouten, and Jasper A. Vrugt. "Probing the limits of predictability: data assimilation of chaotic dynamics in complex food webs." *Ecology letters* 21, no. 1 (2018): 93-103.

We added two new paragraphs to the Discussion section. In the first of them we explain our motivation to use a theoretical model, and how it relates with our research question. In the second one we briefly talk about how our idea can be integrated with field data. The suggested publication is cited in this context.

Minor Comments:

- The title of the paper should indicate that only model simulations are used. e.g. 'Neutral Competition boosts chaos in simulations of food webs'

We followed this advice. The new title is "Neutral competition boosts cycles and chaos in simulated food webs"

- All the results are based on the default parameters listed in Table 1. But what if different values were used for these parameters? Will the final results be sensitive to this? Specifically, will different parameter values produce a less likelihood/probability of chaotic dynamics in the model simulations? The authors should at least discuss this point in context of the final results.

The high number of parameters involved in our family of models makes a systematic exploration of the whole parameter space very difficult. Indeed, controlling the phenomenon we wanted to study with a single parameter was the main design challenge for the numerical experiment.

Nevertheless, we found no differences in the main qualitative results (maximum likelihood of complex dynamics and maximum biodiversity for the neutral case) when our simulations were run under different sets of reasonable parameters. However there are parameter settings possible where chaos and cycles are rare (i.e. when in a 2 dimensional system there are no predator-prey cycles as the prey is less efficient). This will obviously change the result drastically. We chose one of those sets, and stuck to it, for the sake of clarity. Now this is briefly mentioned in the discussion section.

The code provided via Zenodo allows the interested reader to easily re-run our analysis with a set of parameters of his/her choice.

- Section 4, Page 9 Line 3-4: The authors state "These observations suggest..." , yet the authors are referring to model simulations and not observations. I suggest the authors do include a brief discussion on observations, and sort out the vocabulary/terminology used in this section to properly discuss that the results in the paper are from model simulations and not observations.

We fully agree, and scanned the whole document looking for terms that may suggest field observations, and substituted them by more explicit statements.

- Section 4, Page 9 Line 7-8: The authors state "... it may be tempting to conclude that chaos causes diversity." I do not believe this is a valid conclusion, and one might make the opposite conclusion that 'diversity causes chaos'. I recommend the authors be careful when using this type of statement.

This is indeed what we meant, and the paragraph is a warning against that invalid conclusion. We clarified this sentence.

- Is there any way to include uncertainties with the final results? The authors should try to quantify their finding in the abstract/conclusions, e.g. up to 90% (+/- 5%) possibility of chaotic dynamics in the model simulations. A quantified conclusion with uncertainties can be very helpful for readers.

Only figure 4.A, (and derived appendix figure such as A.1 and A.2) lack an explicit uncertainty assessment.

Under the assumption of our sample coming from a multinomial distribution (with categories of stable, cyclic and chaotic) it is easy to calculate standard errors or confidence intervals from the provided information. We used the normal approximation method, that is, estimating the confidence interval as $(p*(1-p)/n)^{0.5}$. Due to the high number of samples ($n = 400$) per competition parameter, none of these confidence intervals was larger than 0.025. Such a confidence interval is too small to be plotted. We can thus consider the ratios to be a good approximation of the actual probabilities.

- Appendix Figures A2, A3, and A5 are not discussed in the main text. This makes it somewhat confusing to understand the role of these figures and what they add to the overall content of the manuscript.

True. We fixed this.

Reviewer: 3

Comments to the Author(s)

GENERAL COMMENTS

The paper explore the effect of competition/neutrality on species diversity and population dynamics. Based on a simple multispecies predator/prey model, the authors explore the probability of chaotic dynamics in response to the average values of the competition matrix. The findings are novel and could inspire a new set of tests and experiments. I suggest to search for and include empirical examples (generated outside the group of investigators) in the line of chaos and biodiversity.

Some minor comments

Introduction Line 18-19- Hubells neutrality requires inmigration for coexistence, as the final result is exclusion of all but one species.

We are not sure of having understood this comment properly. Our model also contains immigration. Moreover our model is not based on the lottery model of Hubbell.

Lines 22-23, also near neutrality does not lead to coexistence sensu-stricto, but it slow downs exclusion. Other mechanisms are required for coexistence.

True. Now we mention it explicitly, although we don't think this is central to the argument.

Line 32- This sentence needs to be justified. Or at least it is not intuitive why should neutral communities are not chaotic. Please expand on this argument as it is central for the rest of the manuscript.

The abovementioned sentence was just an expression of my own surprise for this result. As a physicist, I didn't expect a simpler system to be more prone to chaos... but I was wrong. More specifically, a single-level neutral ecosystem becomes fundamentally one-dimensional, and thus, chaos is ruled out by the Poincaré-Bendixson theorem. We agree that this is cannot be really considered intuitive. We want to keep the mathematical content as low as possible, so we decided to remove this appreciation about intuition in the revised version.

Methods

Please explain better the statistical test and how was it performed.

We added a whole new section to the appendix explaining the Gottwald-Melbourne test and how we used it.

Discussion

Please expand on how to compare this theoretical results with experimental or natural data as to suggest some ways for testing this results..

Also, how common is to found near-neutral competition matrices in nature?

See for example

Fort, H. and Segura, A. 2017. Competition across diverse taxa: quantitative integration of theory and empirical research using global indices of competition. – *Oikos* doi: 10.1111/oik.04756

Or in some empirical estimations of competition matrix
Emergent neutrality drives phytoplankton species coexistence, 2011
PRSCB <https://doi.org/10.1098/rspb.2010.2464>

We added two new paragraphs to the Discussion section. In the first of them we explain our motivation to use a theoretical model, and how it relates with our research question. In the second one we briefly talk about how our idea can be integrated with field data. The suggested literature is cited in this context.

Appendix C

Thank you for considering the review comments and including a detailed reply, as well as including a highlighted version of the manuscript. This made it very easy to see how the manuscript has changed. By addressing my and the other two reviewers' comments, the clarity and quality of the manuscript has increased.

However, two of my original major points (reviewer 1, points 3 and 6) are not addressed satisfactorily, in my view. I therefore still have two major comments, in addition to a few more minor comments. The page numbers mentioned refer to those at the top in the 70-page document which includes the first round of comments, the new manuscript, and another version with highlighted changes.

Major points

1. I still wonder about the influence of the selectivity matrix S on the behaviour of the model (see also my original point 6 (reviewer 1)). Currently, you argue that there is a link between neutral competition and cyclic or chaotic dynamics. However, in the “near-neutral case”, the prey species still experience strong differences because of the highly varying (between 0 and 1) feeding preferences of the predators. Are the prey then really (nearly) neutral in this model when $\varepsilon \approx 0$? In addition, these feeding differences may be solely responsible for creating the cyclic or chaotic dynamics (e.g. compensatory dynamics). It thus seems to me that one could argue that it are the predator feeding preferences creating the cycles or chaos, and not so much the potential near-neutrality of the prey species.

What I wanted to express originally, is that one might be able to investigate the same link between prey neutrality and cyclic or chaotic dynamics in a simpler system consisting of *only one* predator, and multiple prey species. To ensure coexistence of the prey species in this other model structure a Holling-type-III functional response might be necessary (perhaps a Holling-type-II might still work though, depending on how variable the dynamics turn out to be), but there would be no potential confounding influence of predator feeding preferences. My apologies that the original comment was unclear.

In the reply to my original comment you mention following the Dakos et al. 2009 paper also cited in the main text. However, as far as I can tell, this paper does not investigate the effect of prey neutrality at all. I therefore strongly feel that my comment should be addressed in more detail. If you do not agree with my concerns then I would be interested in hearing your response.

Ideally, if you do agree with my concerns, you should argue or show that it is indeed the prey neutrality—and thus not the predator feeding preferences—causing the cycles or chaos, supported by additional simulations. If this cannot be done then at the very least the text should mention something to the effect that the prey species are only neutral from a “bottom-up” perspective, and not so from the predators' point of view.

2. In my original point 3 (reviewer 1), I mentioned the potentially crucial effect that the specific value of the immigration rate f can have on species coexistence. While I

understand that this parameter may be a necessary ingredient of the model, I would be interested in seeing at least an indication that the results obtained are not unique to the specific value $f = 10^{-5}$ chosen. For example, similar to the investigation of the effect of the size of the species pool in Appendix A.1, you could reproduce the different panels of Figure 4 for one higher and one lower f value.

Minor points

3. pg. 37 of 68, l. 48: “preference of different prey species.” *of* should be replaced by *for*.
4. pg. 39 of 68, l. 6: In your reply to my original comment 4 (reviewer 1), you mention that the minimal value of the competition matrix elements is $1 - 0.8 - 0.05 = 0.15$. If I understand everything correctly, would that not imply that the width w of the uniform distribution centered around ϵ is $(\epsilon + 0.05) - (\epsilon - 0.05) = 0.1$ instead of 0.05? If this is the case, please correct this in the main text.
5. pg. 41 of 68, l. 34: Figure 3 shows only the likelihood of chaotic attractors, and not of cyclic ones. I think *cyclic* does not belong here in this sentence as judging from Figure 4A cyclic dynamics appear to be at their least likely for near-neutral competition.
6. Figs. 3, 4, A.1, A.2, A.3, A.5: it could improve clarity for readers scanning through the manuscript to mention at least in the caption that the competition parameter is ϵ , e.g. pg. 43 of 68, l. 28: “... as a function of the competition parameter ϵ .” Alternatively it could be included in all of the axes in the figures.
7. pg. 45 of 68, l. 25: remove “made”
8. Appendix A.2: I enjoyed reading this concise introduction to this method employed for producing the results. One small suggestion which might have reduced some of the initial confusion I had is to consider changing the example timeseries ϕ from $\phi = \{2, 1, 0.5, 0.25, \dots\}$ to e.g. $\phi = \{3, 2, 1.5, 1.25, \dots\}$ which is qualitatively the same but converges to 1 instead of 0 as $k \rightarrow \infty$.

Currently, by having ϕ converge to 0, I do not see how a (pseudo)cyclic path will be generated as the distance to the next step gets smaller and smaller for increasing k . By letting ϕ converge to a non-zero value it is easier to imagine how a (pseudo)cyclic path (which is not a point) might be generated.

The last sentence mentions that the test parameters θ were drawn randomly. Please briefly add whether this means that each timeseries was tested using multiple values of θ , or rather a new θ was drawn for each timeseries.

9. Why is Appendix A.2 referenced first in the main text? It would be more logical to change the order of the appendices.

Appendix D

Associate Editor Comments to Author:

Thank you for the revised paper - while the reviewers observe that you have made efforts to tackle their earlier queries, one of the referees comments that two of their major concerns remain to be satisfactorily answered. Generally, we don't allow multiple rounds of revision, but exceptions can be made where it appears the authors have made efforts to meet the reviewers' concerns, as seems to be broadly the case here. We're going to give you the benefit of the doubt on this occasion, so you can have a final go at revising the paper - but please bear in mind that if the paper doesn't 'get over the line' this time, we won't be able to grant further exceptions. If you've any queries, please contact the editorial office at the appropriate email address, and they'll be glad to assist. Good luck!

We want to say thanks to the editors for this new opportunity. The present version has been expanded with all of the reviewer's suggestions. A file highlighting the differences is provided for your convenience.

Reviewer comments to Author:

Reviewer: 1

Comments to the Author(s)

All comments are in the attached file.

Reviewer: 2

Comments to the Author(s)

Thank you and good job on the revisions.

Reviewer comments

Thank you for considering the review comments and including a detailed reply, as well as including a highlighted version of the manuscript. This made it very easy to see how the manuscript has changed. By addressing my and the other two reviewers' comments, the clarity and quality of the manuscript has increased.

However, two of my original major points (reviewer 1, points 3 and 6) are not addressed satisfactorily, in my view. I therefore still have two major comments, in addition to a few more minor comments. The page numbers mentioned refer to those at the top in the 70-page document which includes the first round of comments, the new manuscript, and another version with highlighted changes.

We want to say thanks to the editor for this rigorous work. The current version of the manuscript has been expanded. All the major points have been addressed. As a result, three new simulations have been added to the manuscript. Only one of the minor points, marked by the reviewer as a suggestion, has been rejected. The reasons are explained below.

Major points

1. I still wonder about the influence of the selectivity matrix S on the behaviour of the model (see also my original point 6 (reviewer 1)). Currently, you argue that there is a link between neutral competition and cyclic or chaotic dynamics. However, in the “near-neutral case”, the prey species still experience strong differences because of the highly varying (between 0 and 1) feeding preferences of the predators. Are the prey then really (nearly) neutral in this model when $\epsilon \approx 0$? In addition, these feeding differences may be solely responsible for creating the cyclic or chaotic dynamics (e.g. compensatory dynamics). It thus seems to me that one could argue that it are the predator feeding preferences creating the cycles or chaos, and not so much the potential near-neutrality of the prey species.

What I wanted to express originally, is that one might be able to investigate the same link between prey neutrality and cyclic or chaotic dynamics in a simpler system consisting of *only one* predator, and multiple prey species. To ensure coexistence of the prey species in this other model structure a Holling-type-III functional response might be necessary (perhaps a Holling-type-II might still work though, depending on how variable the dynamics turn out to be), but there would be no potential confounding influence of predator feeding preferences. My apologies that the original comment was unclear.

In the reply to my original comment you mention following the Dakos et al. 2009 paper also cited in the main text. However, as far as I can tell, this paper does not investigate the effect of prey neutrality at all. I therefore strongly feel that my comment should be addressed in more detail. If you do not agree with my concerns then I would be interested in hearing your response.

Ideally, if you do agree with my concerns, you should argue or show that it is indeed the prey neutrality—and thus not the predator feeding preferences—causing the cycles or chaos, supported by additional simulations. If this cannot be done then at the very

least the text should mention something to the effect that the prey species are only neutral from a “bottom-up” perspective, and not so from the predators’ point of view.

We agree that the chaos and complex cycles is caused by the differences in the preys, in this case by the sensitivity matrix. However our conclusion is that although chaos is created by differences in the sensitivity matrix, it is remarkable that the probability increases so clearly if the competition terms of the prey are neutral. You would expect that the prey species reduce to one species if the competition terms are neutral, but the differences caused by the sensitivity matrix make more complex behavior possible and more likely.

It seems obvious that if all predators and preys are fully neutral (i.e. they all have the same parameters), there cannot be chaos, as the system reduces to a 2 species system. Due to the Poincaré-Bendixson theorem, the most complex behavior that can be found in such a system is cyclic. Similarly there can only be one predator-prey cycle if there is one predator only with neutral preys. Chaos is probably still possible if the competition terms are variable (Van Nes and Scheffer 2004), but this source of chaos is rare and needs highly variable competition terms. We thought that we made this very clear in the text, but we checked again whether our text can be interpreted wrongly.

In order to highlight the fact that non-neutral predation is actually required to observe the effect, we performed an extra run with the parameters of figure 4 (maximum number of species), with the S terms all 0.5 but with random efficiencies of the predation per species. (parameters H random between [1 3] and g randomly between [0.3 0.5]).

In my original point 3 (reviewer 1), I mentioned the potentially crucial effect that the specific value of the immigration rate f can have on species coexistence. While I understand that this parameter may be a necessary ingredient of the model, I would be interested in seeing at least an indication that the results obtained are not unique to the specific value $f = 10^{-5}$ chosen. For example, similar to the investigation of the effect of the size of the species pool in Appendix A.1, you could reproduce the different panels of Figure 4 for one higher and one lower f value.

We added an analysis of the effect of the immigration parameter f to the appendix. Two new values of f were used, 10^{-6} and 10^{-4} (one order of magnitude lower and higher than our standard value) showing that the main qualitative effects remain true.

Minor points

2. pg. 37 of 68, l. 48: “preference of different prey species.” *of* should be replaced by *for*.
Fixed
3. pg. 39 of 68, l. 6: In your reply to my original comment 4 (reviewer 1), you mention that the minimal value of the competition matrix elements is $1 - 0.8 + 0.05 = 0.15$. If I understand everything correctly, would that not imply that the width w of the uniform distribution centered around E is $(E + 0.05) - (E - 0.05) = 0.1$ instead of 0.05? If this is the case, please correct this in the main text.

Fixed.

4. pg. 41 of 68, l. 34: Figure 3 shows only the likelihood of chaotic attractors, and not of cyclic ones. I think *cyclic* does not belong here in this sentence as judging from Figure 4A cyclic dynamics appear to be at their least likely for near-neutral competition.

True. We fixed this.

5. Figs. 3, 4, A.1, A.2, A.3, A.5: it could improve clarity for readers scanning through the manuscript to mention at least in the caption that the competition parameter is ε , e.g. pg. 43 of 68, l. 28: "... as a function of the competition parameter ε ." Alternatively it could be included in all of the axes in the figures.

Done.

6. pg. 45 of 68, l. 25: remove "made"

Done.

7. Appendix A.2: I enjoyed reading this concise introduction to this method employed for producing the results. One small suggestion which might have reduced some of the initial confusion I had is to consider changing the example timeseries φ from $\varphi = \{2, 1, 0.5, 0.25, \dots\}$ to e.g. $\varphi = \{3, 2, 1.5, 1.25, \dots\}$ which is qualitatively the same but converges to 1 instead of 0 as $k \rightarrow \infty$.

Currently, by having φ converge to 0, I do not see how a (pseudo)cyclic path will be generated as the distance to the next step gets smaller and smaller for increasing k . By letting φ converge to a non-zero value it is easier to imagine how a (pseudo)cyclic path (which is not a point) might be generated.

We considered this possibility, but discharged it because it presents some pedagogical challenges. The purpose of this figure is to make a clear link between the Gottwald-Melbourne formula and plane geometry. A cyclic and a chaotic case are shown in the next figure (A.2 in the current submission).

In order to understand the (pseudo)periodicity of those series some familiarity with Fourier analysis is required. As this is just an "in a nutshell" subsection, we rather refer the interested reader to the original paper by Gottwald and Melbourne.

The last sentence mentions that the test parameters θ were drawn randomly. Please briefly add whether this means that each timeseries was tested using multiple values of θ , or rather a new θ was drawn for each timeseries.

We clarified this.

8. Why is Appendix A.2 referenced first in the main text? It would be more logical to change the order of the appendices.

Done.